# Combined Omipalisib and MAPK Inhibition Suppress PDAC Growth

**DOI:** 10.3390/cancers17071152

**Published:** 2025-03-29

**Authors:** Bailey A. Bye, Jarrid L. Jack, Alexandra Pierce, Richard McKinnon Walsh, Austin E. Eades, Prabhakar Chalise, Appolinaire Olou, Michael N. VanSaun

**Affiliations:** 1Department of Cancer Biology, University of Kansas Medical Center, 3901 Rainbow Blvd, Kansas City, KS 66160, USA; 2Department of Biostatistics and Data Science, University of Kansas Medical Center, 3901 Rainbow Blvd, Kansas City, KS 66160, USA

**Keywords:** PDAC, pancreas, pancreatic cancer, therapeutics, targeted therapy, inhibitors, MAPK, PI3K, SHP2

## Abstract

Pancreatic ductal adenocarcinoma (PDAC) is a particularly deadly disease, in part because of a lack of effective therapeutic options. Targeting of mutant KRAS or its effectors is a high priority, as KRAS is mutated at early stages of disease and in almost all PDAC cases. Our study aimed to test dual-targeted therapeutics to suppress two major KRAS effectors: PI3K and MAPK. Dual PI3K/MAPK pathway targeting was more effective than inhibition of either pathway alone when assessing in vitro proliferation, migration, in vivo tumor growth, and survival in mouse models. These data support continued consideration of dual-therapeutic strategies for PDAC.

## 1. Introduction

Pancreatic ductal adenocarcinoma (PDAC) is one of the deadliest cancers, with a 5-year survival rate of 13%, due in part to inherent and acquired resistance to limited therapeutic options [1]. Gemcitabine has been the standard frontline chemotherapeutic for PDAC for more than 20 years but only improves median overall survival by 6–7 months [2,3]. Compared to gemcitabine, more aggressive treatment options, such as FOLFIRINOX, only increase the median overall survival by an additional 4–5 months [3,4]. Effective suppression of PDAC tumor growth with chemotherapy is difficult, partially due to mutations harbored by the cancer cells themselves. KRAS is mutated in over 90% of PDAC, with the G12D mutation being the most common [5,6,7,8]. In a normal cell, KRAS functions as a binary signaling switch by its conversion from an inactive, guanosine diphosphate (GDP)-bound conformation to an active guanosine triphosphate (GTP)-bound conformation, which can then activate multiple downstream signaling cascades [9]. KRAS mutation at G12D impairs GTPase function via the steric hindrance of GTP hydrolysis, which prolongs KRAS in the active GTP-bound state and leads to increased activation of downstream mitogenic pathways [10,11]. The high frequency of mutated KRAS in PDAC has driven interest in developing mutant KRAS-targeted therapeutic strategies. A drug targeting KRAS G12C was approved by the FDA to treat non-small cell lung cancer [12], yet this mutation is only present in less than 3% of PDAC [13]. Generally, many of the other drugs targeting mutated KRAS or its downstream effectors, including ERK, MEK, and AKT, have so far proven ineffective as individual therapeutic targets [6]. While specific drugs targeting mutant KRAS (including KRAS G12D) have shown some promise in treating PDAC, the acquisition of resistance to single-agent therapy remains a concern [14].

The PI3K and MAPK pathways are a central focus for investigation into therapeutic targets for PDAC, as they are known downstream targets of mutant KRAS and play prominent roles in cellular proliferation and survival [15,16,17]. The FDA has approved the usage of a highly potent and specific MEK1/2 (MAPK pathway) inhibitor, Trametinib, in combination with BRAF inhibitor dabrafenib for all non-resectable or metastatic solid tumors (except colorectal cancer), as well as most pediatric gliomas harboring a BRAF V600E mutation [18,19,20,21,22]. MEK inhibitors such as Trametinib have also shown promise as therapeutic strategies for PDAC but display a limited efficacy in clinical trials due in part to the development of resistance [23,24,25,26]. To overcome resistance to MEK inhibition, many current studies are focused on combined therapeutics to block reactivation pathways [27,28]. The PI3K (p110α/β/δ/γ) and mTORC1/2 inhibitor Omipalisib was investigated clinically as a single therapeutic agent in solid tumors, but its efficacy was modest, suggesting that further investigation into an optimal combination treatment strategy is needed [29]. While combined MEK inhibition and PI3K pathway inhibition strategies have shown efficacy in PDAC, as well as other cancers [16,23,30,31,32,33,34,35], they have also faced challenges due to modest effectiveness and therapeutic toxicity [36,37,38,39].

Another strategy for targeting mutant KRAS-driven MAPK activation is by inhibiting KRAS-interacting proteins such as the tyrosine phosphatase SHP2, which is an essential factor in RAS-mediated MAPK signaling [40,41,42]. Recently, SHP2 was shown to be required for mutant KRAS-driven PDAC tumorigenesis [42]. As part of a combination therapeutic strategy, targeting SHP2 via a selective inhibitor such as SHP099 [43] effectively inhibited the MAPK pathway and counteracted therapeutic resistance when combined with a MEK inhibitor in mutant KRAS-driven cancers [44,45,46,47]. Additionally, MAPK inhibition via targeting SHP2 combined with PI3K inhibitors has shown efficacy in multiple types of cancer [48,49,50,51] but remains to be investigated in PDAC.

Our study compared the efficacy of targeting MAPK pathway effectors in PDAC, via SHP099 or Trametinib, when combined with PI3K pathway inhibition, via Omipalisib: either Omipalisib/Trametinib (OmiTram) or Omipalisib/SHP099 (OmiSHP). Our results identified the persistence of PI3K pathway activation upon MAPK pathway inhibition in response to either Trametinib or SHP099 alone. Subsequently, a greater suppression of in vitro tumor cell proliferation and migration, as well as decreased tumor growth in vivo, was observed with dual PI3K/MAPK inhibition.

## 2. Materials and Methods

### 2.1. Cell Culture and Reagents

The K8484 mouse PDAC tumor cell line was derived from the genetic KPC (Pdx-1^cre/+^; LSL-KRAS^G12D/+^; LSL-Trp53^R172H/+^) mouse model of PDAC [52]. The murine PKT62 line was derived from the genetic PKT (Ptf1a^cre/+^; LSL-KRAS^G12D/+^; TGFBR2^flox/flox^) mouse model of PDAC in collaboration with Dr. Nipun B. Merchant and Dr. Nagaraj S. Nagathihalli. MiaPaCa-2 (#CRL-1420) and Panc1 (#CRL-1469) are primary human PDAC tumor cell lines purchased from ATCC (Manassas, VA, USA). All cell lines were maintained in high-glucose (4.5 g/L) Dulbecco’s Modified Eagle’s Medium (Gibco, Grand Island, NY, USA; #1-995-073) supplemented with 5% heat-inactivated fetal bovine serum (Biowest, Bradenton, FL, USA; #S1260) and antibiotic-antimycotic (Gibco, #15240062) at 37 °C with 5% CO_2_. Media was refreshed every 2–3 days. Omipalisib (p110α/β/γ/δ and mTORC1/2 inhibitor; #S2658), Trametinib (MEK1/2 inhibitor, #S2673), and SHP099-HCl (SHP2 inhibitor, #S8278) were purchased from Selleckchem (Houston, TX, USA). For in vitro drug experiments, DMSO controls matched the DMSO concentration in wells containing a combination therapeutic treatment. Unless otherwise specified, drug treatments were diluted to final concentration in DMEM + 5% FBS, and in experiments lasting longer than 48 h, treatment media was refreshed every 2 days. All cell lines were tested for mycoplasma contamination using the InVivoGen MycoStrip™ mycoplasma detection kit (InVivoGen, San Diego, CA, USA; #rep-mys-10) at the conclusion of the study.

### 2.2. Western Blot and Antibodies

Cultured cells were lysed in cold RIPA buffer (Cell Signaling, Danvers, MA, USA; #9806S) supplemented with NaF (10 mM), sonicated, then centrifuged for 10 min at 12,000× *g* at 4 °C. Tumor tissue collected from in vivo studies was homogenized in NaF-supplemented RIPA buffer before sonication and centrifugation at 4 °C. The protein concentration in each lysate was determined using a Pierce BCA protein assay (Thermo Scientific, Rockford, IL, USA; #PI23227). Lysate samples of equal protein concentrations were loaded and run on a 10% SDS-PAGE gel before transfer to nitrocellulose membranes using a Trans-Blot^®^ Turbo™ Transfer System (Bio-Rad, Hercules, CA, USA; #1704150) on a mixed molecular weight setting. Membranes were blocked in TBS with 2% milk and 2% BSA before incubation with the primary antibodies at 4 °C overnight, followed by incubation with appropriate secondary antibodies for 1 h at room temperature. Primary antibodies against phosphorylated AKT (S473, #4060), total AKT (pan, #4298), phosphorylated p44/42 MAPK (Erk1/2, Thr202/Tyr204, #4370), p44/42 MAPK (Erk1/2, #4695), and β-Actin (#CS12262) were purchased from Cell Signaling. The secondary antibody, peroxidase-conjugated anti-rabbit IgG (#711036152), was from Jackson Immunoresearch (West Grove, PA, USA). If applicable, blots were stripped using stripping buffer (Thermo Scientific, #PI34577) for at least 30 min at room temperature and were checked for sufficient stripping before addition of the next primary antibody. Pico or femto substrate (Thermo Scientific #PI34577 or #PI34095) was added to all blots prior to imaging using either a Fluorchem M (Bio-Techne, Minneapolis, MN, USA; 92-15312-00) or Azure 600 (Azure Biosystems, Dublin, CA, USA; AZI600-01) imaging system.

### 2.3. EdU Proliferation Assay

Measuring cell proliferation via 5-ethynyl-2′-deoxyuridine (EdU) incorporation has been previously described [53]. Briefly, cells were seeded at 50,000 cells/well in a 24-well plate and allowed to attach overnight. The next morning, drug treatments were applied to respective wells. After 24 h, wells were spiked with EdU (Invitrogen, Eugene, OR, USA; #A10044) and allowed to incubate for 6 h. Cells were then detached from the plate with trypsin (Gibco, #25-200-056) and fixed overnight on a rocker at 4 °C in 5% buffered formalin. After fixing, cells were permeabilized and incubated with copper sulfate (1 mM), azide dye (AF647, Invitrogen, #AI0277, 2 µM), and sodium ascorbate (100 mM) and stained with PI (0.5 µg/mL) (Invitrogen, #P3566) before analysis via flow cytometry. Cells were gated for singlets, then PI positivity, then EdU positivity. Cell proliferation was reported as the percentage of EdU+ cells within the total PI+ cell population, using negative control cells (not incubated with EdU) to define the gating for analysis.

### 2.4. Migration Assay

Cell migration assays were conducted using the Incucyte^®^ Live Cell Analysis System. Cells were first seeded on an Incucyte^®^ Imagelock 96-well plate (Sartorius, Bohemia, NY, USA; #BA-04855) in a confluent layer and allowed to attach overnight. The next morning, cells were incubated with Mitomycin C (5 µg/mL) (Thermo Scientific, #AAJ67460XF) for 2 h to halt proliferation. Using the Incucyte^®^ 96-Well Woundmaker Tool (Sartorius, #4563), a scratch was created across the center of each well. After rinsing wells with sterile PBS, drug treatment media was added, and the plate was placed in the Incucyte^®^ incubator for analysis. An image of each well was acquired every two hours. Wound width analysis was performed using the Incucyte^®^ Scratch Wound Analysis Software Module (Sartorius, #9600-0012, version 2023A Rev2) over 24 h, starting 2 h after the plate was placed in the incubator. Wells displaying insufficient cell density outside of the scratch, that were out of focus, or in which the analysis definition failed to accurately define the wound area were excluded from the final analysis.

### 2.5. Colony Formation Assay

Cells were seeded at 1000 cells/well and allowed to attach overnight. Treatment media was then added to each well and was refreshed every two days. At the endpoint, cells were stained with crystal violet solution (Sigma-Aldrich, St. Louis, MO, USA; #V5265, 1%) for 20 min and rinsed with water. The crystal violet precipitate was dissolved in 10% acetic acid, and the absorbance for each well was read at 590 nm.

### 2.6. Histology

Tissues were fixed immediately after collection in 10% formalin at room temperature overnight and transferred to 70% ethanol before paraffin processing, embedding, and sectioning. Slides were deparaffinized with HistoClear (Electron Microscopy Sciences, Hatfield, PA, USA; #64110) and rehydrated using an ethanol series. For H&E staining, slides were stained with hematoxylin and eosin (Electron Microscopy Sciences #26252-01 and #26252-02), then dehydrated and mounted in Permount mounting medium (Thermo Fisher #SP15-100). For immunofluorescence staining, heat-mediated antigen retrieval was performed using citrate buffer (10 mM), and slides were blocked in blocking buffer containing 5% donkey serum for 1 h at room temperature. Slides were incubated with primary antibody overnight at 4 °C before washing and incubation with secondary antibody and Hoechst nuclear stain (Thermo Fisher #H3570, 5 µg/mL) for one hour at room temperature. Cells were mounted in ProLong™ Diamond Antifade Mountant (Thermo Fisher #P36965) before imaging. Primary antibodies: Collagen Type I (Rockland #600-401-103-0.1, 1:200, Pottstown, PA, USA), Ki67 (Abcam #ab15580, 1:500, Waltham, MA, USA), Claudin 18 (Proteintech #66167-1-Ig, 1:500, Rosemont, IL, USA), and IBA1 (FUJIFILM Wako #019-19741, Richmond, VA, USA). Secondary antibodies: Donkey anti-Mouse IgG, Alexa Fluor™ 488 (Thermo Fisher #A-21202, 1:1000) and Donkey anti-Rabbit IgG, Alexa Fluor™ 594 (Thermo Fisher #A-21207, 1:1000). The Trichrome stain was performed by the University of Kanas Cancer Center’s Biospecimen Repository Core Facility (BRCF) staff. All slides were imaged using an EVOS M5000 (Thermo Fisher #AMF5000) microscope. For tumor area quantification (H&E), slides were scanned using a Pathscan Enabler IV slide scanner (Meyer Instruments, Houston, TX, USA). Images were processed to quantify normal pancreas, tumor, necrotic, and other/unspecified tissue areas using Adobe Photoshop. For the Trichrome stain quantification: 2–3 fields of view were collected in non-necrotic, cell-rich areas of tissue, and the blue fibrotic area was measured using Adobe Photoshop and ImageJ (version 1.54d). For immunofluorescence staining quantification: for Collagen I and Claudin 18/Ki67, 8 fields of view at 20× magnification were collected and quantified for each tumor. For IBA1, 3 fields of view were collected and quantified. Visual fields were collected randomly for the Collagen I and IBA1-stained slides and via Claudin18 positivity for the Claudin18/Ki67-stained slides. Percent of the Collagen I+, Claudin 18+, or IBA1+ area was calculated with ImageJ. Ki67+ nuclei were counted manually to ensure the exclusion of non-Claudin 18+ cells and any non-specific, cytoplasmic Ki67 staining. Immunofluorescence quantification is represented as the average for each tumor. Samples from mice that unexpectedly succumbed to disease, resulting in non-viable tissue, were excluded from histological analysis. If images were adjusted for quantification or representation (i.e., brightness/contrast or color thresholding), all pixels across the entire image were adjusted equally, with care taken to represent the data accurately.

### 2.7. Mice

Animal procedures and maintenance used for this research were conducted in accordance with the University of Kansas Medical Center IACUC guidelines (approved from 2019 to 2025) to ensure humane and ethical treatment. In accordance with the approved IACUC guidelines, subcutaneous tumors were established by injecting 1 × 10^6^ tumor cells into the flank of wild-type C57BL/6 mice (JAX:000664). Both male and female mice were used in our studies. Subcutaneous tumor size was monitored every 2 days using calipers. Dosing for both sets of in vivo experiments, the subcutaneous model or PKT (Ptf1a^cre/+^; LSL-KRAS^G12D/+^; TGFBR2^flox/flox^) spontaneous PDAC model, utilized respective drugs that were dissolved in 0.5% methylcellulose with >0.05% Tween80 and administered via oral gavage 3 times per week (1 mg/kg Trametinib, 0.3 mg/kg Omipalisib, and 50 mg/kg SHP099). Treatment for mice with subcutaneous tumors began when the tumors reached an approximate volume of 100 mm^3^ and then treatment continued for either 18 days or until the tumors reached an endpoint size of 2000 mm^3^. In the PKT mice, drug treatment began at 4.5 weeks of age and continued for 10 weeks or until the humane endpoint was reached.

### 2.8. Seahorse Extracellular Flux Analysis

For the Glycolysis Stress Test assays, on the day before the experiment, K8484 cells were plated at 40,000 cells per well in normal growth medium (5% FBS DMEM), and a Seahorse XFe96 sensor cartridge (Agilent, Santa Clara, CA, USA; #103793-100) was incubated in a 96-well plate containing 200 μL/well of XF Calibrant Solution at 37 °C in a non-CO_2_ incubator overnight. On the day of the experiment, Seahorse XF DMEM medium (Agilent, #103575-100) was supplemented to a final concentration of 2 mM glutamine (Gibco™, #25030081) and 1 mM sodium pyruvate (Gibco™, #11360070). The media on the cells was replaced with 180 μL/well of the prepared assay media ± drug treatments and incubated at 37 °C for 1 h (Vehicle: 1:5k DMSO, Omipalisib 25 nM, SHP099 20 μM, and Trametinib 10 nM). During this time, injections were prepared in the assay medium without drug treatments. Port A: 18 μL of 100 mM Glucose (Gibco™, #A2494001) and 100 μg/mL Hoechst (Invitrogen, #H3570); Port B: 20 μL of 10 μM Oligomycin (Sigma-Aldrich, #O4876-5MG); and Port C: 22 μL of 1M 2-DG (Acros, Geel, Belgium; #111980010), pH adjusted to 7.4. The sensor cartridge was loaded into a Seahorse XFe96 and calibrated. The plate containing cells was loaded into the instrument following calibration and run according to the manufacturer-provided protocol for Glycolysis Stress Tests. Briefly, 3 cycles of 3 min of alternating mixing and measuring were performed prior to the first injection and following each injection. The cells were counted following conclusion of the assay using the BioTek Cytation 1 imaging platform (Agilent Technologies, Inc., BioTek Cytation 1 Cell Imaging Multi-Mode Reader, RRID:SCR_019730, Santa Clara, CA, USA), and ECAR values were reported per 1000 cells for analysis. In the event of biologically impossible results or recorded errors in plate preparation, whole columns (as loaded) were removed from the data analysis, as described in [54].

### 2.9. Statistical Analysis

For the proliferation assays, 4 replicate wells were analyzed per experimental treatment in all 4 cell lines; for the migration assays, 4–6 replicate wells were analyzed per experimental treatment in all 4 cell lines; for the colony formation assays, 3 replicate wells were analyzed per experimental treatment in all 4 cell lines. For the subcutaneous tumor experiments, vehicle: n = 5 (16 days) and 4 (18 days); Omipalisib: n = 6; Trametinib: n = 6; SHP099: n = 5; OmiTram: n = 5; OmiSHP: n = 5. Tumor growth among the six experimental groups of mice was measured at baseline and every two days for 18 days (0, 2, 4, 6, 8, 10, 12, 14, 16, and 18). Differences in tumor growth over time among the groups were analyzed using linear mixed models for repeated measure data using SAS (version 9.4) procedure GLIMMIX (SAS Institute Inc.) using an AR(1) covariance structure. For tumor growth between groups over time, interactions were assessed by specifying appropriate contrast statements within the modeling framework. For the in vivo experiments using PKT mice, Vehicle: n = 6; Omipalisib: n = 6; Trametinib: n = 6; OmiTram: n = 7. Survival analysis included all listed mice, but all other post-mortem analyses excluded mice that unexpectedly succumbed to disease before final weights and tissues could be collected (post-mortem analysis: vehicle: n = 5; Omipalisib: n = 6; Trametinib: n = 5; OmiTram n = 6). In all analyses except that of tumor growth rate in the in vivo subcutaneous tumor experiments, statistical significance was assessed by one-way ANOVA with Tukey’s multiple comparison analysis using GraphPad Prism 5 software (* *p* < 0.05, ** *p* < 0.01, *** *p* < 0.001, and **** *p* < 0.0001). Mice in all in vivo experiments were randomly assigned to each experimental group. Since the study was exploratory, no formal sample size calculation and power analysis were carried out to determine the sample size. The number of mice in the study, as well as sex distribution across treatment groups, was based on the availability of mice for the study.

## 3. Results

### 3.1. Inhibition of MAPK or PI3K Signaling Maintains Alternative Mitogenic Signaling Pathway Activation

The tumor cell persistence in response to therapeutic drugs is due in part to their ability to activate alternative tumorigenic signaling pathways when dominant mitogenic pathways are suppressed [55]. The PI3K and MAPK pathways are two prominent mitogenic pathways driving pancreatic ductal adenocarcinoma (PDAC), in part due to the constitutive activation of KRAS, which is mutated in almost all PDAC [5,6,7,8]. Therefore, we aimed to assess the activation status of each pathway after treatment with targeted therapeutics. We measured PDAC tumor cell response to drugs targeting the PI3K or MAPK pathways using Omipalisib (PI3K pathway inhibitor), Trametinib (MAPK pathway inhibitor via ERK1/2), and SHP099 (MAPK pathway inhibitor via SHP2). Effective drug concentrations were determined by phospho-specific detection of target proteins via dose–response curves (Appendix A).

First, we assessed pathway action after PI3K pathway inhibition via Omipalisib in KRAS mutant mouse (K8484 and PKT62) and human (MiaPaCa-2 and Panc1) PDAC cell lines. Omipalisib treatment demonstrated effective inhibition of AKT phosphorylation (pAKT) at 5 nM (K8484) and 25 nM (PKT62, MiaPaCa-2, and Panc1). Additionally, we observed sustained MAPK activation as measured by phosphorylated ERK (pERK) levels in response to the Omipalisib treatment (Figure 1 and Appendix A).

Next, we assessed MAPK pathway inhibition via pERK in response to either Trametinib or SHP099 treatment. Treatment with MEK inhibitor Trametinib demonstrated effective pERK suppression at 10 nM (K8484 and PKT62) or 20 nM (MiaPaCa-2 and Panc1) (Figure 1 and Appendix A). Alternatively, inhibition of SHP2 via SHP099 suppressed pERK at 20 μM (PKT62, MiaPaCa-2, and Panc1) or 50 μM (K8484) (Figure 1 and Appendix A). We also noted sustained PI3K pathway activation (as measured by pAKT) in response to MAPK pathway inhibition via Trametinib or SHP099 in all cell lines.

In combination treatment groups, Trametinib appeared to be more effective than SHP099 in suppressing pERK overall, yet the K8484 cell line demonstrated relative resistance to pERK inhibition via Trametinib when combined with Omipalisib. Omipalisib relatively inhibited pAKT in all cell lines for both individual, as well as combination, treatments when compared to either Trametinib or SHP099 alone, though the MiaPaCa-2 cell line sustained relatively more AKT phosphorylation in the OmiTram group than the other cell lines. Overall, these results indicate that PDAC cells require inhibitors of both PI3K and MAPK pathways to achieve optimal mitogenic signaling suppression by both pathways.

### 3.2. Combined Targeting of PI3K and MAPK Pathways Inhibits Proliferation, Colony-Forming Ability, and Migration of PDAC Cells In Vitro

To assess the effect of PI3K/MAPK inhibition on PDAC tumor cell growth and aggressiveness in vitro, we tested the in vitro effects of OmiTram or OmiSHP treatment on PDAC tumor cell proliferation, colony-forming ability, and migration. First, we assessed the effect of OmiTram or OmiSHP treatment on cell proliferation via the EdU incorporation assay (Figure 2A,B). Combination treatment with OmiTram more effectively suppressed cell proliferation than either Omipalisib or Trametinib treatment alone in all cell lines tested (Figure 2A). Combination treatment with OmiSHP also most effectively suppressed proliferation compared to the vehicle control group in all cell lines, and OmiSHP treatment was more effective than either Omipalisib or SHP099 treatment alone in two of the four cell lines tested (K8484 and Panc1) (Figure 2B).

We further tested the effect of each combination therapeutic on tumor cell survival and growth via colony formation assay (Figure 2C,D). The OmiTram dual therapeutic more effectively suppressed colony formation in the Panc1 cell line than either Omipalisib or Trametinib treatment alone (Figure 2C and Appendix A). Treatment with OmiSHP more effectively suppressed colony formation in the MiaPaCa-2 cell line compared to either Omipalisib or SHP099 treatment alone (Figure 2D and Appendix A).

We next measured the effectiveness of each combination treatment strategy on the PDAC cell migratory ability using a scratch wound-healing assay (Figure 3, Appendix A). The OmiTram combination treatment more effectively inhibited migration in three of the four cell lines tested (K8484, PKT62, and MiaPaCa-2) compared to individual treatments alone (Figure 3A), and the OmiSHP combination treatment more effectively inhibited migration in only the MiaPaCa-2 cell line than either single-agent treatment (Figure 3B). Notably, among all the single-agent treatments, individual Omipalisib treatment most effectively inhibited migration in all cell lines compared to Trametinib or SHP099, which is consistent with other studies reporting inhibited tumor cell migratory capacity upon PI3K pathway inhibition [56,57,58].

Overall, the combined inhibition of PI3K and MAPK pathways via Omipalisib/Trametinib or Omipalisib/SHP099 treatment in vitro suppresses PDAC tumor cell proliferation, migration, and colony-forming ability in multiple PDAC tumor cell lines.

### 3.3. Effects of PI3K and MAPK Pathway Inhibition on Tumor Metabolism

Constitutive RAS signaling as a result of mutant KRAS in PDAC has been implicated in promoting rapid cell proliferation via increasing glycolytic flux, which is likely dependent on downstream MAPK (but not PI3K) signaling [59]. To briefly interrogate the combined effect of PI3K and MAPK inhibition on tumor cell metabolism, we subjected K8484 cells to a Seahorse Glycolysis Stress Test (Appendix A). Only direct inhibition of MEK with Trametinib incited a significant reduction in the basal Extracellular Acidification Rate (ECAR) following glucose addition, which was interpreted as the glycolysis rate (Appendix A) but this was not recapitulated in the OmiTram group. However, when Oligomycin was added to halt the mitochondrial use of pyruvate to obtain the maximal glycolytic capacity, all therapeutic treatments except for Omipalisib or Trametinib alone significantly lowered the glycolytic capacity of the cells (Appendix A). Notably, the SHP099 and OmiSHP treatments were the only groups with significantly reduced glycolytic reserve (Appendix A). This suggests that glucose may be preferentially used for aerobic glycolysis to lactic acid fermentation in SHP2-inhibited cells rather than mitochondrial respiration. Nevertheless, neither the OmiSHP nor OmiTram combination therapeutic altered tumor cell metabolism more significantly than both single-agent treatments individually.

### 3.4. Combined PI3K and MAPK Pathway Inhibition Suppresses Tumor Growth In Vivo

We next assessed the efficacy of a combined PI3K/MAPK-targeted therapeutic strategy in vivo. To monitor the effect of combination PI3K and MAPK inhibition on tumor growth in vivo over time, we injected KPC PDAC tumor cells (K8484) subcutaneously into the flank of C57BL/6 mice. After the tumors reached a sufficient starting volume, mice were treated 3 times per week via oral gavage with either vehicle control (0.5% methylcellulose), Omipalisib (0.3 mg/kg), Trametinib (1 mg/kg), SHP099 (50 mg/kg), combined Omipalisib/Trametinib (OmiTram), or combined Omipalisib/SHP099 (OmiSHP). The tumor volume was measured every 2 days for 18 days. One mouse in the vehicle group reached the humane endpoint before the end of the 18-day period (at day 16). Over the 18-day study period, the tumor growth rate of the Trametinib, OmiTram, and OmiSHP groups was significantly suppressed compared to the control group (Figure 4A). However, while both the OmiSHP and OmiTram-treated mice displayed no net overall tumor growth increase for the first 10 days of the study, the OmiSHP group displayed increased tumor growth after 10 days, which trended upon greater overall tumor growth compared to the OmiTram group by the 18-day study period endpoint (*p* = 0.06) (Figure 4A, File S2). At day 18, the OmiTram-treated mice represented the only group with both significantly reduced tumor volume and weight at the endpoint compared to the vehicle control group (Figure 4A–C). Together, these data indicate that, while combined PI3K and MAPK inhibition via OmiSHP or OmiTram slows tumor growth in vivo, the OmiTram combined therapeutic most effectively inhibited tumor growth over time.

Since the OmiTram treatment demonstrated greatest efficacy in the implanted tumor model, we further assessed whether OmiTram administration would also suppress tumor growth and progression in an aggressive spontaneous mouse model of PDAC. The PKT (Ptf1a^cre/+^; LSL-KRAS^G12D/+^; TGFBR2^flox/flox^) mouse model typically exhibits PanIN formation at 3.5 weeks of age and reaches the endpoint at 7–10 weeks of age [60]. PKT mice received the vehicle control (0.5% methylcellulose), Omipalisib (0.3 mg/kg), Trametinib (1 mg/kg), or OmiTram treatment 3 times per week starting at 4.5 weeks of age and continuing until a humane endpoint was reached or 10 weeks of treatment was completed (mice surpassed 14 weeks of age, representing a doubling of the median survival). While all mice in the Omipalisib and Trametinib individual treatment groups reached the humane endpoint before the experimental endpoint, five out of seven mice treated with the OmiTram combination therapeutic completed 10 weeks of treatment and surpassed 14 weeks of age (Figure 5A). Tumor weight normalized to total body weight at the endpoint was significantly lower in the OmiTram group than in the vehicle, Omipalisib, and Trametinib individual treatment groups (Figure 5B and Appendix A), and no visible metastatic outgrowth in the liver, lungs, or peritoneal cavity was observed. Additionally, the pancreata from the OmiTram group displayed significantly less tumor area than the Omipalisib or Trametinib groups compared to the vehicle control group (Figure 5C).

We further performed a general characterization of the tumor stroma that included IF staining for macrophages and fibrotic content. IBA1 has been used by others to identify macrophage populations in PDAC [61,62,63,64]. A slight decrease in macrophage infiltration was observed for the OmiTram group when compared to either the vehicle or the Trametinib-treated groups but not for the Omipalisib single treatment group (Appendix A). When assessing the extracellular matrix content, we observed no significant difference in fibrosis as measured by Collagen I between the treatment groups at the endpoint (Figure 5D) (similarly to what we observed in the implanted tumors from Figure 4 after day 18 (Appendix A)).

When probing for tumor cell proliferation via Ki67 expression in the endpoint tumors, we found that all treatment groups exhibited relatively low levels of Ki67, except for the single-agent Trametinib-treated group, which displayed increased Ki67 staining despite the increased survival time compared to the vehicle and Omipalisib-treated groups (Figure 5E). We speculate that this may be due to the development of resistance in the single agent Trametinib tumors, which appears to have been mitigated by the combination treatment with Omipalisib (OmiTram). Overall, these data indicate that simultaneous targeting of PI3K and MAPK signaling pathways via Omipalisib and Trametinib more effectively suppresses tumor growth in the PKT spontaneous mouse model of PDAC than treatment with either drug individually.

## 4. Discussion

Our study has demonstrated sustained or upregulated PI3K pathway activation upon MAPK pathway inhibition with Trametinib (MEK1/2 inhibitor) or SHP099 (SHP2 inhibitor), as well as sustained or upregulated MAPK pathway activation upon PI3K pathway inhibition via Omipalisib (p110α/β/δ/γ and mTORC1/2 inhibitor), in PDAC tumor cell lines. These results support others that show the upregulation of MAPK activation upon the inhibition of various components of the PI3K-AKT-mTOR pathway in solid cancers and vice versa [23,32,65]. Inversely, others have shown that PI3K inhibition alone induces concurrent MAPK pathway inhibition, rather than compensatory activation, in cancer [17,66,67]. Consequent MAPK pathway inhibition upon PI3K inhibition has also been demonstrated in PDAC tumor cells but only in the presence of wild-type or inactivated KRAS [17]. Since the vast majority of PDAC presents with oncogenic mutant KRAS [5,6,7,8], dual therapeutics targeting PI3K and direct inhibitors of mutant KRAS [13,68] may prove more effective than therapeutics targeting downstream targets of mutant RAS, such as MEK [17,23]. Furthermore, while the activation of certain PI3K class I isoforms can be stimulated by RAS (p110α, δ, and γ), others may be activated by G-coupled protein receptor signaling (p110β and γ) [69]. Therefore, strategic, and perhaps isoform-specific, targeting of the PI3K pathway combined with mutant KRAS or related MAPK pathway effector inhibition may prove more effective in suppressing PDAC than either targeted therapeutic strategy alone.

Our study also demonstrated that Omipalisib application more effectively suppressed tumor cell migration than MAPK pathway inhibition via Trametinib or SHP099 (Figure 3). Furthermore, treatment with combined Omipalisib/Trametinib was more effective at inhibiting migration than treatment with either drug alone in three out of the four PDAC cell lines tested (Figure 3A). Genetic and pharmacological PI3K and mTOR inhibition have been shown by others to suppress migratory and invasive activity in PDAC [56,57,58]. Our results additionally suggest that including PI3K pathway inhibitors as part of a treatment strategy targeting the MAPK pathway may prove more effective at inhibiting multiple aspects of PDAC tumor progression (i.e., proliferation and migration) than single pathway-targeting agents alone.

In addition to our experiments demonstrating anti-tumor effects of combined PI3K/MAPK inhibition, we have highlighted a novel implication for these inhibitors in regulating tumor metabolism. Notably, the addition of Omipalisib to both Trametinib and SHP099 treatments in the OmiTram and OmiSHP groups did not change the glycolytic capacity or glycolytic reserve compared to either Trametinib or SHP099 treatment alone (Appendix A). Additionally, while MAPK pathway inhibition via Trametinib (Tram and OmiTram) resulted in a significant decrease in glycolytic capacity compared to the vehicle and Omipalisib alone (Appendix A), SHP099 treatment (in SHP099 and OmiSHP groups) induced the greatest reduction in glycolytic capacity and was the only treatment that significantly inhibited the glycolytic reserve (Appendix A). The difference in metabolic perturbation mediated by MEK1/2 or SHP2 inhibition, coupled with the apparent resistance to SHP099 in the K8484 cell line both in vitro and in vivo (Figure 1 and Figure 4), suggests a potential MAPK-independent role for SHP2 in regulating PDAC tumor cell metabolism. Therefore, other therapeutic agents, such as those targeting metabolic pathways upon which SHP099-treated cells exhibit dependency, may also demonstrate enhanced efficacy as part of a combination therapeutic strategy in PDAC.

Lastly, two different studies have shown that, in preclinical models, combined MAPK and PI3K inhibition enhanced the PDAC tumor response to gemcitabine and gemcitabine/nab-paclitaxel, the current standard chemotherapeutic strategies for PDAC [36,37]. However, the effect was modest and, in the case of combined PI3K/MAPK inhibition combined with gemcitabine alone, this treatment strategy did not show more effectiveness than the clinically approved combined gemcitabine/erlotinib treatment [37]. Toxicity has also been identified as a major consideration in combination therapeutic strategies involving MAPK and PI3K inhibition. Though the dual inhibition of MEK and PI3K has shown promise in preclinical models, it has not been well tolerated in clinical trials involving patients with advanced solid tumors, including PDAC [38,39]. For example, a phase 1b clinical trial in advanced solid tumors using a combined Omipalisib/Trametinib strategy demonstrated some anti-tumor efficacy but was ultimately terminated due to toxicity in the patients [39]. Though SHP2 inhibitors are being investigated in clinical trials (NCT03114319 and NCT05354843) and SHP2 inhibition has been preliminarily shown to counteract therapeutic resistance to MEK inhibition [44], further investigation into whether a SHP2/PI3K-targeted dual therapeutic strategy would demonstrate clinical efficacy might be considered.

## 5. Conclusions

Altogether, our results demonstrate that MAPK pathway inhibition via Trametinib or SHP099 combined with PI3K pathway inhibition via Omipalisib act together to inhibit PDAC tumor cell growth and migration more effectively than either single therapeutic agent alone. While an Omipalisib/Trametinib therapeutic strategy was more effective than an Omipalisib/SHP099 therapeutic in our murine PDAC study, a clinical trial has already demonstrated the significant toxicity of a PI3K/mTOR and MEK inhibitor combination that prevents further clinical development [39]. Nevertheless, our data support targeting both these pathways; we believe that further technological developments are needed to develop tumor-selective or targeted delivery to limit adverse side effects and still achieve the optimal combination therapeutic strategies to target PI3K and MAPK pathways in tandem to treat PDAC.

## Figures and Tables

**Figure 1 cancers-17-01152-f001:**
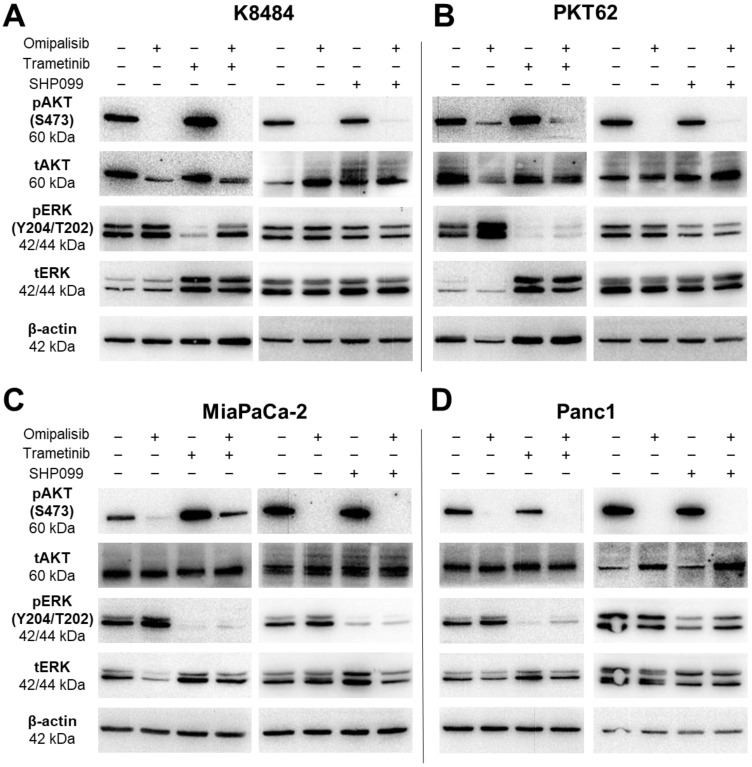
Inhibition of MAPK or PI3K signaling maintains alternative mitogenic signaling pathway activation. Western blots demonstrating inhibition of either PI3K (pAKT compared to the tAKT levels) or MAPK (pERK compared to the tERK levels) as well as combined inhibition of pathways upon targeted therapeutic treatment (24 h for treatments containing Omipalisib and/or Trametinib or 3 h for treatments containing SHP099). (**A**) K8484 cells were treated with 5 nM Omipalisib, 10 nM Trametinib, and/or 50 μM SHP099. (**B**) PKT62 cells were treated with 25 nM Omipalisib, 10 nM Trametinib, and/or 20 μM SHP099. (**C**) MiaPaCa2 and (**D**) Panc1 cells were treated with 25 nM Omipalisib, 20 nM Trametinib, and/or 20 μM SHP099. The original Western blot figures can be found in Appendix A.

**Figure 2 cancers-17-01152-f002:**
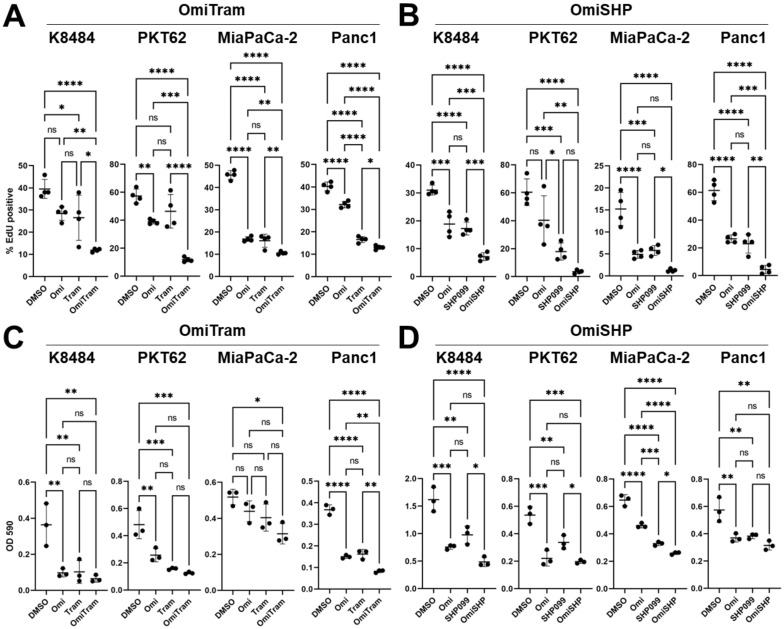
Combined targeting of PI3K and MAPK pathways inhibits proliferation and colony forming ability of PDAC cells in vitro. Changes in cell proliferation in response to either OmiTram (**A**) or OmiSHP (**B**) treatment for 24 h was measured via the EdU assay. K8484 cells were treated with 5 nM Omipalisib, 10 nM Trametinib, and/or 50 μM SHP099. PKT62 cells were treated with 25 nM Omipalisib, 10 nM Trametinib, and/or 20 μM SHP099. MiaPaCa-2 cells were treated with 25 nM Omipalisib, 1 nM Trametinib, and/or 20 μM SHP099. Panc1 cells were treated with 25 nM Omipalisib, 1 nM Trametinib, and 20 μM SHP099. n = 4 wells for all groups. (* *p* < 0.05, ** *p* < 0.01, *** *p* < 0.001, and **** *p* < 0.0001). Changes in colony-forming ability in response to either OmiTram (**C**) or OmiSHP (**D**) treatment was measured via absorbance after crystal violet stain dissolution. K8484 cells were treated with 5 nM Omipalisib, 10 nM Trametinib, and/or 20 μM SHP099. PKT62 cells were treated with 5 nM Omipalisib, 10 nM Trametinib, and/or 2 μM SHP099. MiaPaCa2 cells were treated with 5 nM Omipalisib, 5 nM Trametinib, and/or 20 μM SHP099. Panc1 cells were treated with 50 nM Omipalisib, 20 nM Trametinib, and/or 50 μM SHP099. n = 3 wells for all groups (ns = not significant (*p* ≥ 0.05), * *p* < 0.05, ** *p* <0.01, *** *p* < 0.001, and **** *p* < 0.0001). Error bars are shown ±SD.

**Figure 3 cancers-17-01152-f003:**
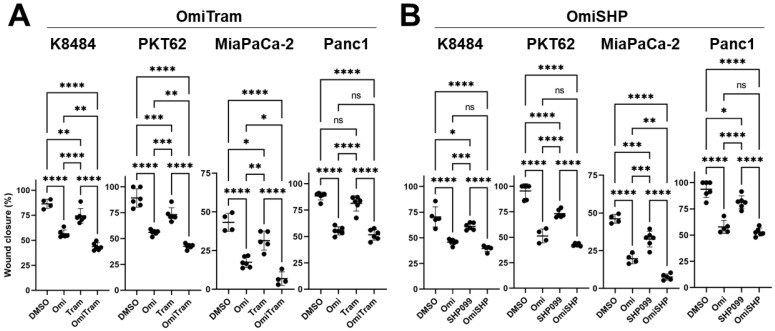
Combined targeting of PI3K and MAPK pathways inhibits the migratory ability of PDAC cells in vitro. Changes in the migratory ability after treatment with either OmiTram (**A**) or OmiSHP (**B**) was assessed via the scratch migration assay over 24 h. K8484 cells were treated with 5 nM Omipalisib, 5 nM Trametinib, and/or 50 μM of SHP099. PKT62 cells were treated with 10 nM Omipalisib, 10 nM Trametinib, and/or 50 μM of SHP099. MiaPaCa2 cells were treated with 25 nM Omipalisib, 20 nM Trametinib, and/or 20 μM SHP099. Panc1 cells were treated with 5 nM Omipalisib, 20 nM Trametinib, and/or 20 μM SHP099. n = 4–6 wells for all groups (ns = not significant (*p* ≥ 0.05), * *p* < 0.05, ** *p* <0.01, *** *p* < 0.001, and **** *p* < 0.0001). Error bars are shown ±SD.

**Figure 4 cancers-17-01152-f004:**
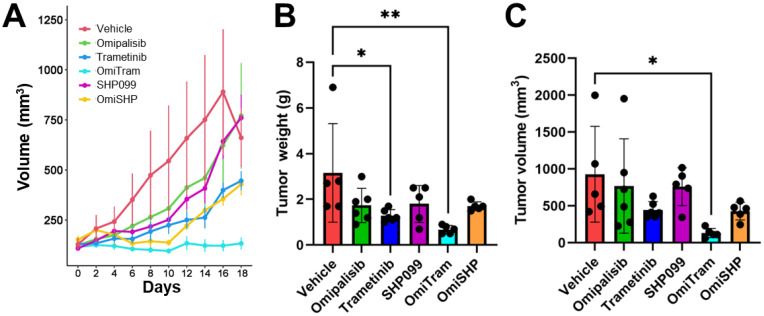
Combined targeting of PI3K and MAPK pathways suppresses tumor growth in an in vivo subcutaneous model of PDAC (K8484 cells). (**A**) Mean tumor growth rate over time of mice treated with Omipalisib, Trametinib, SHP099, OmiTram, or OmiSHP. Error bars are shown as ±standard error. (**B**) Tumor weight analysis after 18 days of treatment. (**C**) Endpoint tumor volume after 18 days of treatment. Error bars are shown ±SD. Vehicle: n = 5 (16 days) or 4 (18 days); Omipalisib: n = 6; Trametinib: n = 6; SHP099, n = 5; OmiTram, n = 5; OmiSHP, n = 5 (* *p* < 0.05, ** *p* < 0.01).

**Figure 5 cancers-17-01152-f005:**
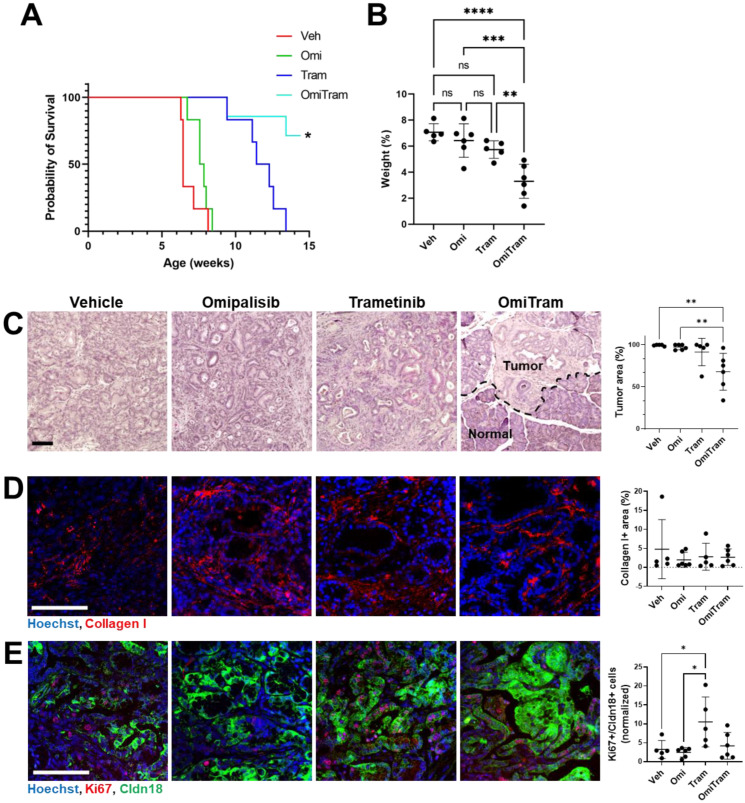
Combined targeting of the PI3K and MAPK pathways suppresses tumor growth in an in vivo spontaneous mouse model of PDAC (PKT GEM model, Ptf1a^cre/+^; LSL-KRAS^G12D/+^; TGFBR2^flox/flox^). (**A**) Survival analysis of mice treated with the vehicle, Omipalisib, Trametinib, or combined OmiTram-targeted therapeutic. “*” represents the study endpoint (10 weeks after start of treatment). Vehicle: n = 6; Omipalisib: n = 6; Trametinib: n = 6; OmiTram: n = 7. (**B**) Endpoint tumor weight was measured as a percentage of the total mouse weight. (**C**) Representative 10× H&E images of the endpoint tumors. The percent of the tumor area was measured from the entire scanned H&E-stained tissue sections. (**D**) Representative images of Collagen I staining in the endpoint PKT tumors. (Blue: Hoechst, red/pink: Collagen I). Quantification represents the average percent Collagen I-positive area per 20× field of view (n = 8 random fields per tumor). (**E**) Representative images of Ki67 and Claudin 18 staining in the endpoint PKT tumors. (Blue: Hoechst, red/pink: Ki67, green: Cldn18). Quantification represents the average number of Ki67+ nuclei in Claudin 18-positive cells per 20× field of view (n = 8 fields of view, identified by positive Claudin 18 staining). Quantifications were normalized to the percent of Claudin 18+ area per field. Post-mortem analysis: vehicle: n = 5; Omipalisib: n = 6; Trametinib: n = 5; OmiTram n = 6 (* *p* < 0.05, ** *p* < 0.01, *** *p* < 0.001, and **** *p* < 0.0001). Scale bars = 200 μm. Error bars show ±SD.

## Data Availability

The data generated from this study are available from the corresponding author upon reasonable request.

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
