# Peer review of "Combined Omipalisib and MAPK Inhibition Suppress PDAC Growth"

_cancers, 2025, doi:10.3390/cancers17071152_

Round 1

Reviewer 1 Report

Comments and Suggestions for Authors

Comments to the Author:

The general experimental design is fine. The written is good as well. Several questions as follow:

1. Had the 5% FBS been heat-inactivated in every cell culture medium? How and why? (Line 111)

2. When and how many times were the cells tested for mycoplasma contamination? (Line 119)

3. The authors haven’t specified the centrifugation temperature. (Line 123)

4. How many males and females, respectively, were included in this study? Did both sexes respond in the same manner to inhibitors? (Line 208)

5. Would the authors consider using less significant markers to avoid confusion (P Ë‚ 0.001 and P Ë‚ 0.0001 look redundant)? (Line 261)

6.  The significant maker P hadn’t been shown in italic format in the whole manuscript.

7. Regarding the Western blot data, could the authors enhance the quality of the bands? (The three bubble bands in Figure 1D, p-AKT in K8484 in Figure S1A, β-actin in K8484 and PKT62 in Figure S1B, tERK in PKT62 in Figure S1C).

8. Have the authors tracked the drug’s inhibitory effect over time to capture any unexpected relapse of expression?

9. SHP099 appears to exhibit insufficient inhibition of the MAPK pathway in K8484 across all examined doses, which could significantly affect subsequent experiments. (Figure S1C)

10. The cells (K8484) seem to behave very differently in response to omipalisib, when comparing Figure S2A and S2B. In Figure S2A, the majority of cells died in omipalisib. While in Figure S2B, only a small number of cells died in omipalisib. Such inconsistent observations can be very misleading.

11.  In Figure S2A, it seems that the Panc1 cells were either seeded in too small amounts, making it difficult to evaluate the drug inhibition effect, or the cells failed to grow even in the DMSO control well. Did authors test this with multiple replicates? Could the authors provide a rationale?

12.  In Figure S2B, it seems that Panc1 cells were seeded unevenly in all four wells resulting in a highly concentrated cell mass in the center or the formation of double layers, which might have undermined the efficacy of the treatments.

13. In Figure 4A, why was the tumor volume reduced from day 16 to day 18 in the vehicle group?

14. In Figure 4B, both combined treatments (OmiTram and OmiSHP) didn’t further inhibit tumor growth when compared to single-agent treatment, which didn’t support the authors’ conclusion that “combined PI3K inhibition and MAPK inhibition synergizes to suppress PDAC”.

15.  In Figure 5B, the tumor weights were expressed with inconsistent unit (%) when compared to Figure 4B (g). Why? Did these two expression manners yield different results?

16. In Figure 5E, besides the abnormally high expression of Ki67 in single-agent trametinib treatment, the combined treatment (OmiTram) shows an increase in the cell proliferation marker (Ki67) compared to single-agent Omipalisib treatment. Could this be an artifact or is there another underlying cause? Could the authors provide an explanation?

17. As shown in Figure S4, the combined treatments (OmiTram and OmiSHP) did not further inhibit tumor cell migration compared to Omi treatment alone in all four cell lines. These images do not seem to support the authors' conclusion.

18. Considering all the experimental models used in this project were KRASG12D-driven PDAC, it may be beneficial to include this phenotypic information in the title of manuscript to avoid confusion.

19. Would the authors think, treatments with trametinib and SHP099 against KrasG12D PDAC could have stronger side effects on healthy tissues compared to a KRASG12D-targeted drug (e.g., MRTX1133 has been cleared for clinical evaluation by FDA)?

Author Response

We would like to thank the reviewer for their attention to detail and help in improving this manuscript.

Comment 1. Had the 5% FBS been heat-inactivated in every cell culture medium? How and why? (Line 111)

Response 1: Yes, heat inactivation of FBS is a routine laboratory practice and included in most established cell culture protocols. Our laboratory uses sterile-filtered, heat-inactivated FBS (Biowest #S1480) for all cell culture applications. Biowest performs heat inactivation by heating the FBS in a water bath at 65 °C for 30 minutes. Heat inactivation of FBS has been shown to destroy heat-labile proteins, such as complement components found in serum, and some microorganisms, which can interfere with cell growth. Though we have not specifically tested whether FBS heat inactivation is necessary in the context of the experiments performed in this study, the same heat-inactivated FBS from Biowest was consistently used in all cell culture maintenance and experiments reported in this manuscript.

Comment 2. When and how many times were the cells tested for mycoplasma contamination? (Line 119)

Response 2. Cells were tested for mycoplasma contamination upon initiation of the project and at the conclusion of this study. We have updated these details in the Methods section (line 118-120).

Comment 3. The authors haven’t specified the centrifugation temperature. (Line 123)

Response 3. All lysates were centrifuged at 4°C; we have clarified this in the Methods section (lines 123-125).

Comment 4. How many males and females, respectively, were included in this study? Did both sexes respond in the same manner to inhibitors? (Line 208)

Response 4. Sex distribution across treatment groups in both sets of in vivo experiments was as follows:

Subcutaneous tumors (Figure 4)
Vehicle: 2F, 3M
Ompalisib: 2F, 4M
SHP099: 2F, 4M
Trametinib: 2F, 3M
OmiSHP: 5M
OmiTram: 5M

PKT GEM model (Figure 5)
Vehicle: 4F, 2M
Omipalisib: 3F, 3M
Trametinib: 2F, 4M
OmiTram: 3F, 4M

As stated in the Materials and Methods section under Statistical Analysis (Line 271-272), sex distribution of mice across treatment groups was based on availability of mice for this study. Though we cannot comment on the OmiSHP and OmiTram groups in Figure 4 since they made up of only male mice, there was no notable difference in response to inhibitors between sexes in the other groups. In the PKT GEM model experiment described in Figure 5, there was no notable difference in survival time or endpoint tumor weight associated with sex.

Comment 5. Would the authors consider using less significant markers to avoid confusion (PË‚ 0.001 andP Ë‚ 0.0001 look redundant)? (Line 261)

Response 5. We agree with the reviewer that both P Ë‚ 0.001 and P Ë‚ 0.0001 both indicate significant differences, and that distinguishing between the two may result in needless confusion. However, we also acknowledge that retaining the significance markers we have used may better illustrate the differences between groups (such as the differences between Omi/OmiTram (P < 0.001) and Veh/OmiTram (P < 0.0001) in the PKT62 cell line Figure 2A. Though we don’t specifically describe these details in the manuscript text, we think that including the significance markers as we’ve delineated them may allow for more detailed interpretation of the data if desired by the reader.

Comment 6. The significant maker P hadn’t been shown in italic format in the whole manuscript.

Response 6. We thank the reviewer for pointing this out and have fixed this in the manuscript text.

Comment 7. Regarding the Western blot data, could the authors enhance the quality of the bands? (The three bubble bands in Figure 1D, p-AKT in K8484 in Figure S1A, β-actin in K8484 and PKT62 in Figure S1B, tERK in PKT62 in Figure S1C).

Response 7. We agree that aesthetics are important, but for the dose-response Western blots in Figure S1—the original lysates were collected in 2020, so they were no longer suitable for Western blot analysis. Unfortunately, with the short turn-around time we did not have sufficient time to re-run all the lysates. However, the outcome will likely be the same, as it is only visual aesthetics that would change. We would like to clarify that the purpose of Figure S1 was to provide a guideline for experimental doses.

Comment 8. Have the authors tracked the drug’s inhibitory effect over time to capture any unexpected relapse of expression?

Response 8. This question is somewhat unclear, if the reviewer is asking about potential resistance mechanisms through long term exposure, then no we have not done an expression analysis of resistant cells. We have monitored the duration of drug activity after administration to cells in culture via phosphorylation of target proteins and that is how we designed our treatment strategies.

Comment 9. SHP099 appears to exhibit insufficient inhibition of the MAPK pathway in K8484 across all examined doses, which could significantly affect subsequent experiments. (Figure S1C).

Response 9. We agree with the reviewer and confirm that the K8484 cell line shows less suppression of pERK via SHP099 compared to the other cell lines. However, this dose is functionally effective if you look at the proliferation data, suggesting potential alternate mechanisms for SHP099 (non-MAPK activity) or different sensitivity based on KRAS mutation; i.e. G12D (K8484, PKT62, Panc1) vs G12C (MiaPaCa-2).

Comment 10. The cells (K8484) seem to behave very differently in response to omipalisib, when comparing Figure S2A and S2B. In Figure S2A, the majority of cells died in omipalisib. While in Figure S2B, only a small number of cells died in omipalisib. Such inconsistent observations can be very misleading.

Response 10. We will try to clarify the confusion. In Figure S2A and S2B, each row is representative of one experimental plate (for example, a 12-well plate where each cell line was plated in triplicate). Assay endpoint was determined for each individual plate via colony size/density in the DMSO wells, and the entire plate was imaged and quantified at once. Since endpoint was determined per plate via colony size/density in the DMSO wells, each experimental treatment group should only be compared to the DMSO controls in each given treatment group (i.e. Figure 2C and S2A should not be compared to Figure 2D and S2B). The observation of higher colony density in the K8484 Omipalisib wells in S2A compared to S2B could be due to slightly different experimental endpoints, and should only be compared to other wells within its respective experimental group.

Comment 11. In Figure S2A, it seems that the Panc1 cells were either seeded in too small amounts, making it difficult to evaluate the drug inhibition effect, or the cells failed to grow even in the DMSO control well. Did authors test this with multiple replicates? Could the authors provide a rationale?

Response 11. We do agree with the reviewer and think that a longer assay timepoint may provide better results for the Panc1 OmiTram experiment shown in S2A. We re-ran the assay and replaced the figure in S2A and the quantification in Figure 2C. Though the assay was repeated, the interpretation of the data remains the same. As explained in point 10, each cell line per treatment group was assessed at the same endpoint, but the experimental endpoint may have differed among the same cell line in different treatment groups (i.e. in S2A compared to S2B).

Comment 12. In Figure S2B, it seems that Panc1 cells were seeded unevenly in all four wells resulting in a highly concentrated cell mass in the center or the formation of double layers, which might have undermined the efficacy of the treatments.

Response 12. The observation is understandable. Unfortunately, that cell line likes to clump more than the others. Even with clumps, the result of the experiment is still consistent with the conclusion that the combination treatment was effective and the drugs still reduce the clumped cells.

Comment 13. In Figure 4A, why was the tumor volume reduced from day 16 to day 18 in the vehicle group?

Response 13. Tumor volume decreases after day 16 in the Vehicle group because one mouse was euthanized due to the volume of its tumor exceeding the 2000 mm3  endpoint value as detailed in the methods section. We have added clarification in the manuscript text (lines 396-397, and line 424).

Comment 14. In Figure 4B, both combined treatments (OmiTram and OmiSHP) didn’t further inhibit tumor growth when compared to single-agent treatment, which didn’t support the authors’ conclusion that “combined PI3K inhibition and MAPK inhibition synergizes to suppress PDAC”.

Response 14. We do agree that if one considers endpoint tumor weight alone in Figure 4B, the tumors from the OmiTram group did not weigh significantly less than those from any of the non-Vehicle treatment groups. Additionally, we acknowledge that both the Trametinib and OmiTram-treated groups also showed significantly reduced tumor weight at endpoint compared to the Vehicle group. However, OmiTram was the only treatment that exhibited no tumor growth as measured over time (Figure 4A, File S1). Additionally, the OmiTram group was the only group with significantly reduced tumor volume compared to Vehicle at the 18-day experimental endpoint (Figure 4C (added)). The combined differences in tumor growth rate (Figure 4A) and endpoint tumor weight and volume (Figure 4B, C) led us to conclude that combined PI3K and MAPK inhibition via OmiTram more effectively suppressed tumor growth than single-agent treatment in this experiment.

Comment 15. In Figure 5B, the tumor weights were expressed with inconsistent unit (%) when compared to Figure 4B (g). Why? Did these two expression manners yield different results?

Response 15. We understand the reviewer’s confusion and have clarified these results. We added endpoint tumor weight in grams as well as mouse weight at endpoint in Figure S6B and C. In short, we noticed similar trends among the two tumor weight measurements (% and g). We chose to report the endpoint tumor weight as a percentage of mouse weight due to the fact that the PKT mice develop tumors between 6-8 weeks of age, before they have reached their full adult body weight. Therefore, the Vehicle and Omipalisib-treated mice reached the humane endpoint when they were younger and smaller than the mice that survived for longer (i.e. Trametinib and OmiTram groups) and had more time to grow.

Comment 16. In Figure 5E, besides the abnormally high expression of Ki67 in single-agent trametinib treatment, the combined treatment (OmiTram) shows an increase in the cell proliferation marker (Ki67) compared to single-agent Omipalisib treatment. Could this be an artifact or is there another underlying cause? Could the authors provide an explanation?

Response 16. We agree this isn’t exactly what we expected. We believe that the high Ki67 expression is accurate as we’ve reported it. While a couple of the mice in the OmiTram-treated group showed potentially higher Ki67 expression in tumor cells than the mice in the Omipalisib group, the difference was not statistically significant. We could speculate upon multiple hypotheses as to why this is, but this would require additional experimentation and is outside of the scope of this study.

Comment 17. As shown in Figure S4, the combined treatments (OmiTram and OmiSHP) did not further inhibit tumor cell migration compared to Omi treatment alone in all four cell lines. These images do not seem to support the authors' conclusion.

Response 17. We would like to clarify that the images depicted in S4 are representative images of the quantified data in Figure 3: where the OmiTram treatment inhibited migration more effectively than single-agent therapeutics in all cell lines except for Panc1, where it was not more effective than the Omipalisib alone. As discussed in the Results (line

 349-356), the OmiSHP combination treatment did not inhibit tumor cell migration more effectively than Omipalisib alone, except for in the MiaPaCa-2 cell line. The representative image chosen in Figure S4 was blinded by choosing the image closest to the “average measurement” for each treatment shown in Figure 3, so as not to bias the representation. The graphs in Figure 3 contain all of the replicates, which is supportive of our conclusions.

Comment 18. Considering all the experimental models used in this project were KRASG12D-driven PDAC, it may be beneficial to include this phenotypic information in the title of manuscript to avoid confusion.

Response 18. This is not completely accurate. Only three of the four cell lines represent KRASG12D-driven PDAC, with the exception of the MiaPaCa-2 cell line, which is driven by KRASG12C. Since over 90% of PDAC presents with a KRAS mutation, and the G12D mutation is most common, we don’t think this title is misleading as the project is also a combined analysis of in vitro and in vivo data.

Comment 19. Would the authors think, treatments with trametinib and SHP099 against KrasG12DPDAC could have stronger side effects on healthy tissues compared to a KRASG12D-targeted drug (e.g., MRTX1133 has been cleared for clinical evaluation by FDA)?

Response 19. Yes, our combination is likely to have more side effects than if we used a G12D-specific inhibitor. Newer studies using MRTX1133 would be ideal, but we do not currently have this drug in the lab.

Reviewer 2 Report

Comments and Suggestions for Authors

The manuscript aims to study the effects of a combination of inhibitors targeting the phosphoinositide 3-kinase (PI3K)/Akt and mitogen-activated protein kinase (MAPK) signaling pathways in cellular and tumor models of pancreatic ductal adenocarcinoma (PDAC). The materials and methods are well described. To evaluate the combined effect of the drugs on in vitro models, modern approaches were used, including a clonogenic test, a migration test, an EdU proliferation assay, and a Seahorse extracellular flux analysis. Additionally, the study presents the results of in vivo experiments using a spontaneous PDAC mice model.

As inhibitors of PI3K/Akt and MAPK signaling pathways, the authors used Omipalisib (p110α/β/δ/γ and mTORC1/2 inhibitor) in combination with two different MAPK signaling pathway inhibitors: Trametinib (MEK1/2 inhibitor) or SHP099-HCL (SHP099; SHP2 inhibitor). As a result, the authors concluded that the therapeutic strategy based on the Omipalisib/Trametinib combination was more effective than the Omipalisib/SHP099 combination in their study on a mouse PDAC model. However, it has already been demonstrated that the combination of PI3K/mTOR and MEK inhibitors has significant toxicity, which limits its further clinical development. Thus, despite the effective combined action of the inhibitors, the authors conclude that further technological developments are needed for the selective and targeted delivery of drugs to tumors to avoid side effects of the drugs.

The manuscript mentions the synergistic effect of signaling pathway inhibitors, and the title states «Combined PI3K inhibition and MAPK inhibition synergizes to suppress PDAC». To be sure that a synergistic effect is present, appropriate calculation methods need to be used (Duarte et al., 2022; Tallarida, 2011). Therefore, it is strongly recommended to use these calculation approaches to prove the synergistic effect. Otherwise, one should talk about the combined or general effect of inhibitors.

It is difficult to evaluate the results of Western blot due to the absence of quantitative analysis of the obtained data in the article. For example:

line 283-285: Additionally, we observed sustained or upregulated MAPK activation as measured by phosphorylated ERK (pERK) levels (Figure 1 and Figure S1a).

The level of phosphorylated ERK in Western blot is difficult to determine. It is essential to calculate the results and present statistics, as done for other experiments. Additionally, in this sentence, it is essential to specify in which cell lines and under the influence of which inhibitor changes in phosphorylated ERK level are observed.

Line 290-292: We also noted sustained or upregulated PI3K pathway activation (as measured by pAKT) in response to MAPK pathway inhibition via Trametinib or SHP099 in all cell lines.

Similarly, it is difficult to determine the level of phosphorylated AKT in cell lines without proper data processing. For example, using SHP099 (Figure S1c, PKT62, and MiaPaCa-2 cell lines), the level of phosphorylated and total AKT decreases proportionally with increasing concentration of inhibitor SHP099, yet these changes are challenging to assess without quantitative analysis.

Figure 1A, using the Western blot of K8484 cell line lysates as an example

When comparing the two membranes (membrane 1: upper left, wells 1 and 2; membrane 2: upper right, wells 1 and 2), the signals for phosphorylated and total forms of AKT, as well as phosphorylated and total forms of ERK, should be relatively identical. However, there is a significant discrepancy between them, while these are the same control wells.

To clarify the details, was DMSO used as a solvent control for the cells prior to lysis and subsequent Western blotting?

The interpretation of the results of the colony-forming assay remains unclear. For example:

Line 337-339: The OmiTram dual therapeutic more effectively suppressed colony formation in MiaPaCa-2 and Panc1 cell lines compared to vehicle-treated control cells than either Omipalisib or Trametinib treatment alone (Figure 2c, Figure S2a).

According to Figure 2c, there is no difference in the effect of OmiTram in the MiaPaCa-2 cell line compared to the individual treatments with Omipalisib and Trametinib.

Line 340-342: Treatment with OmiSHP more effectively suppressed colony formation compared to vehicle-treated controls in the K8484 and MiaPaCa-2 cell lines compared to either Omipalisib or SHP099 treatment alone (Figure 2d, Figure S2b).

According to Figure 2d, OmiSHP also shows an effect that differs from SHP099, but not from Omipalisib in the PKT62 cell line, as in the K8484 cell line. The results do not mention that OmiSHP also effectively suppressed colony formation in the PKT62 cell line.

Recommendations to in vivo experiments (Figure 4):

It is recommended the authors combine the tumor growth graphs since it is difficult to compare them separately to assess the effects of individual inhibitors and their combinations.

It is also recommended to compare tumor volume at each time point in addition to tumor weight comparison to evaluate the effect of different treatments.

Highly recommended to estimate the treatment response according to Response Evaluation Criteria in Solid Tumors version 1.1 (RECIST1.1).

Therefore, even this study is actual and has a good methodological foundation, it requires re-interpretation and recalculation of the data to ensure accurate conclusions.

Author Response

We would like to thank the reviewer for their thoughtful comments and help in improving this manuscript.

Comment 1: The manuscript mentions the synergistic effect of signaling pathway inhibitors, and the title states «Combined PI3K inhibition and MAPK inhibition synergizes to suppress PDAC». To be sure that a synergistic effect is present, appropriate calculation methods need to be used (Duarte et al., 2022; Tallarida, 2011). Therefore, it is strongly recommended to use these calculation approaches to prove the synergistic effect. Otherwise, one should talk about the combined or general effect of inhibitors.

Response 1: We have changed the wording in the paper to remove statements claiming synergistic effects between the two drugs in the absence of the proper calculation approaches and molecular analysis to prove “synergy”. That would be beyond the scope of this project.

Comment 2: It is difficult to evaluate the results of Western blot due to the absence of quantitative analysis of the obtained data in the article. For example:

Response 2: We understand that reviewers like to see quantification. However, the western blots are not meant to be quantitative nor described as quantitative for this study. The drugs have previously been shown to be selective for their targets in the dose ranges we have used. The importance of our observation is in the trends across the treatments and the different cell lines and to demonstrate that targeting on pathway alone does not reduce the other. We’ve re-worded our interpretation in the Results section to reflect this.

Comment 3: line 283-285: Additionally, we observed sustained or upregulated MAPK activation as measured by phosphorylated ERK (pERK) levels (Figure 1 and Figure S1a).

The level of phosphorylated ERK in Western blot is difficult to determine. It is essential to calculate the results and present statistics, as done for other experiments. Additionally, in this sentence, it is essential to specify in which cell lines and under the influence of which inhibitor changes in phosphorylated ERK level are observed.

Response 3: We agree—we have clarified this in the Results section (lines 289-292) and removed “upregulated,” as that was not consistently seen.

“Additionally, we observed sustained MAPK activation as measured by phosphorylated ERK (pERK) levels in response to the Omipalisib treatment (Figure 1 and Figure S1a).”

Comment 4: Line 290-292: We also noted sustained or upregulated PI3K pathway activation (as measured by pAKT) in response to MAPK pathway inhibition via Trametinib or SHP099 in all cell lines.

Response 4: We agree and have reworded this in the Results section (lines 297-299): “We also noted sustained PI3K pathway activation (as measured by pAKT) in response to MAPK pathway inhibition via Trametinib or SHP099 in all cell lines.”

Comment 5: Similarly, it is difficult to determine the level of phosphorylated AKT in cell lines without proper data processing. For example, using SHP099 (Figure S1c, PKT62, and MiaPaCa-2 cell lines), the level of phosphorylated and total AKT decreases proportionally with increasing concentration of inhibitor SHP099, yet these changes are challenging to assess without quantitative analysis.

Response 5: This is an interesting observation that the total may be lost; although it is difficult to ascertain the physiological impact of total loss versus phospho loss. Ideally, we should have checked more downstream pathways to see if this was perpetuated and it would be a really interesting follow-up study. As such, data from figure 1 shows that at the concentration of SHP099 used, the pAKT persists similar to vehicle.

Comment 6: Figure 1A, using the Western blot of K8484 cell line lysates as an example

When comparing the two membranes (membrane 1: upper left, wells 1 and 2; membrane 2: upper right, wells 1 and 2), the signals for phosphorylated and total forms of AKT, as well as phosphorylated and total forms of ERK, should be relatively identical. However, there is a significant discrepancy between them, while these are the same control wells.

Response 6: We agree with the reviewer with the fact that the signal for phosphorylated and total AKT and ERK should ideally be similar across the OmiSHP and OmiTram groups, especially in the Vehicle control group. However, it is important to note that the left and right sides of each part of this figure (i.e. left and right side of figure 1A, for example) represent two independent experiments at two different time points (24h vs 3h) on two separate membranes. Therefore, small differences in the experiment runs as well as technical differences such as exposure of the Western blot could account for the differences seen between the groups.

Comment 7: To clarify the details, was DMSO used as a solvent control for the cells prior to lysis and subsequent Western blotting?

Response 7: Yes, DMSO was used as a solvent control in all in vitro experiments (DMSO concentration corresponds with the combination treatment wells, as they contain the most DMSO relative to the other treatments).

Comment 8: The interpretation of the results of the colony-forming assay remains unclear. For example:

Line 337-339: The OmiTram dual therapeutic more effectively suppressed colony formation in MiaPaCa-2 and Panc1 cell lines compared to vehicle-treated control cells than either Omipalisib or Trametinib treatment alone (Figure 2c, Figure S2a).

According to Figure 2c, there is no difference in the effect of OmiTram in the MiaPaCa-2 cell line compared to the individual treatments with Omipalisib and Trametinib.

Line 340-342: Treatment with OmiSHP more effectively suppressed colony formation compared to vehicle-treated controls in the K8484 and MiaPaCa-2 cell lines compared to either Omipalisib or SHP099 treatment alone (Figure 2d, Figure S2b).

According to Figure 2d, OmiSHP also shows an effect that differs from SHP099, but not from Omipalisib in the PKT62 cell line, as in the K8484 cell line. The results do not mention that OmiSHP also effectively suppressed colony formation in the PKT62 cell line.

Response 8: We thank the reviewer for pointing this out and have clarified in the text—this error in reporting the results in the K8484 and PKT62 cell lines was a result of over-interpretation of p values on our end and was incorrect as stated in the Results text. We believe the interpretation of the colony-forming assay described in Figure 2C, D is now more accurate in the Results section (lines 343-348).

Comment 9: Recommendations to in vivo experiments (Figure 4):

It is recommended the authors combine the tumor growth graphs since it is difficult to compare them separately to assess the effects of individual inhibitors and their combinations.

Response 9: We agree and originally had it this way until asked to separate them by a different reviewer. We have replaced the separated tumor volume graphs with one combined graph to facilitate interpretation of the data in Figure 4.

Comment 10: It is also recommended to compare tumor volume at each time point in addition to tumor weight comparison to evaluate the effect of different treatments.

Response 10: Our primary readout for drug effectiveness in this assay was the rate of tumor growth over time, which takes into account tumor volume at each time point (see File S1). In response to another reviewer comment, we have also added a direct comparison of endpoint tumor volume to Figure 4 (Figure 4C).

Comment 11: Highly recommended to estimate the treatment response according to Response Evaluation Criteria in Solid Tumors version 1.1 (RECIST1.1).

Response 11: Great recommendation, but one study was not performed in coordination with a clinical trial and we do not want to mislead the readers.

Therefore, even this study is actual and has a good methodological foundation, it requires re-interpretation and recalculation of the data to ensure accurate conclusions.

Reviewer 3 Report

Comments and Suggestions for Authors

Dear Author,

The article "Combined PI3K inhibition and MAPK inhibition synergizes to 2 suppress PDAC" described the experimental proof of advancing PDAC treatment employing PI3K and MAPK inhibition, resulting in reduced resistance. However, I have a few suggestions to improve the understanding and quality of this work

Comments/Suggestion

1) Several studies have evaluated the effect of PI3K and MAPK inhibition on PDAC. How does this study vary from those?

2) Given the availability of KRAS G12D targeting drugs, is there a reason not to employ them in this study?

3) Lane 117. Is it 5% FBS or 10% FBS?

4) Is there a specific reason to use Edu proliferation assay instead of the simpler MTT or SRB assay?

5) Do you have western blots for MEK1/2?

6) In Figure 1A, Omipalisib treatment reduced AKT in the left side blot compared to the control. However, on the right side panel, tAKT is increasing more than control. Is this possible with the same drug and cell line? Is it due of the short time frame? Why can't you attempt the same time point -24 hours?

7) Fig. 1A. Right panel, SHP099 does not significantly reduce pERK, nor does the combination? Is there any reason?

8) Figure 1a shows that the combination of omipalisib and trametinib leads to resistance by raising p-ERK. Could you please run a densitometry analysis and provide the value. From loading control western, it appears that you somewhat overloaded the protein.

9) Figure 1B shows that omipalisb treatment raises p-Erk in the left panel but not in the right panel. Why are these things happening? Is it due of the short time frame similar to Fig 1A?

10) Figure 4a. Is there any reason why the tumor volume decreases after day 16 in Vehicle control?

11) Figure 5: Why you didn’t use SHP099 and Omni+shp099 for the in vivo study. Efficacy may vary during in vivo experiments. If you have data, put it in the supplementary section.

12) Figure 5. It would be good to show changes in the tumor microenvironment after treatment. At the very least, alterations in CD8 ( you can also show CD4, macrophages, NK cells, MDSCs etc.)

13) Can you show the exact tumor volume or weight graph in Figure 5B and the animal weight graph in the supplementary?

14) Figure 5D shows no alterations in the stroma, which may be attributable to the PKT GEM model. Can you provide a similar staining (trichrome or Sirius red stain) for the in vivo study using the KPC cell line? Because KPC tumors have more stroma which resemble human PDAC?

15. Did you check the pERK and p-MEK levels in the in vivo tumor after therapy to ensure that the pathways were inhibited?

Best Wishes

Author Response

We appreciate the time and effort this reviewer has put into providing insightful comments that have improved this manuscript.

Comments/Suggestion

Comment 1. Several studies have evaluated the effect of PI3K and MAPK inhibition on PDAC. How does this study vary from those?

Response 1. We have modified the title to help in this clarification. Additionally, our study does support others evaluating the effect of combined PI3K and MAPK inhibition on PDAC (references 16,23, 30-35). The novel aspect of our study is the evaluation of the effectiveness of MAPK pathway inhibition targeting SHP2 in combination with the PI3K inhibitor Omipalisib; which has been investigated in other cancers (references 48-51) but not yet in PDAC.

Additionally, PI3K (p110α/β/δ/γ) and mTORC1/2 inhibition via Omipalisib has not yet been studied in combination with MAPK inhibition via Trametinib specifically in PDAC tumors. A phase 1b clinical trial in advanced solid tumors using a combined Omipalisib/Trametinib strategy (reference) displayed some anti-tumor efficacy, but included only 7 PDAC tumors.

Comment 2. Given the availability of KRAS G12D targeting drugs, is there a reason not to employ them in this study?

Response 2. Yes these studies should be done; it is unfortunately out of the scope and funding of this project. A previous study (reference 17) showed concurrent inhibition, rather than apparent compensatory activation, of the MAPK pathway upon PI3K inhibition in PDAC cells with wild-type or inactivated KRAS. Therefore, we think it may be reasonable to hypothesize that using an inhibitor of mutant KRAS (instead of a MEK inhibitor like Trametinib) may prove more effective in combination with a PI3K inhibitor at suppressing both MAPK and PI3K pathway activity, as we discussed briefly (lines 462-467). Additionally, this study was performed before the wide availability of KRAS G12D-targeting drugs, but we think expanding upon our results using a mutant or pan-KRAS targeted drug in a separate study could address concerns about development of resistance to single-agent KRAS-targeted therapy while reducing toxicities associated with Trametinib and other related MAPK inhibitors.

Comment 3. Lane 117. Is it 5% FBS or 10% FBS?

Response 3. All in vitro experiments were performed in DMEM + 5% FBS.

Comment 4. Is there a specific reason to use Edu proliferation assay instead of the simpler MTT or SRB assay?

Response 4. The EdU assay measures cells that are actively replicating by incorporating the modified Uridine into the DNA during S phase. Since MAPK and/or PI3K can contribute to cell cycle, we wanted to assess actual cellular replication. The MTT assay is a metabolic assay, as such mutant KRAS is known to alter cellular metabolism and therefore is a different readout. It can be used but interpreted differently and it does not necessarily implicate replication. SRB could be used as a total cell number count, this could be used but we do not use this assay in the lab.

Comment 5. Do you have western blots for MEK1/2?

Response 5. Unfortunately no, and we did not have enough time to perform these in the allotted resubmission timeframe. We decided to measure functional MEK inhibition via Trametinib with ERK phosphorylation, as it is directly downstream of MEK.

Comment 6. In Figure 1A, Omipalisib treatment reduced AKT in the left side blot compared to the control. However, on the right side panel, tAKT is increasing more than control. Is this possible with the same drug and cell line? Is it due of the short time frame? Why can't you attempt the same time point -24 hours?

Response 6. When conducting preliminary dose-response assays using SHP099 (Figure S1), we found that that MAPK inhibition via SHP099 was most effective at the three hour timepoint. Therefore, to best assess the relative effects of MAPK inhibition via SHP099 on the PI3K pathway, we decided to lyse the cells after 3 hours, rather than 24. The different timepoint may account for the differences in tAKT levels between the left (24 hours) and right (3 hours) panels in Figure 1A.

Comment 7. Fig. 1A. Right panel, SHP099 does not significantly reduce pERK, nor does the combination? Is there any reason?

Response 7. We agree with the reviewer and confirm that the K8484 cell line shows less suppression of pERK via SHP099 compared to the other cell lines. However, this dose is functionally effective considering the proliferation data, suggesting potential alternate mechanisms for SHP099 (non-MAPK activity) or different sensitivity based on KRAS mutation (i.e. G12D (K8484, PKT62, Panc1) vs G12C (MiaPaCa-2)).

Comment 8. Figure 1a shows that the combination of omipalisib and trametinib leads to resistance by raising p-ERK. Could you please run a densitometry analysis and provide the value. From loading control western, it appears that you somewhat overloaded the protein.

Response 8. This is an understandable comment. All Western blot gels were loaded with the same amount of protein per well as described in the Methods section (lines 126-129). The concept of this analysis was to not over-interpret the signal for one particular band, but to demonstrate relative trends across cell lines for pERK and pAKT, noting that single agents don’t suppress the other pathway. We have reworded section 3.1 describing Figure 1 to better represent the information conveyed with the Western blots (lines 287-315) and de-emphasize any upregulation as we did not intend to investigate mechanisms behind this potential effect.

Comment 9. Figure 1B shows that omipalisb treatment raises p-Erk in the left panel but not in the right panel. Why are these things happening? Is it due of the short time frame similar to Fig 1A?

Response 9. It is possible that the apparent differences in pERK levels could be due to the shorter time frame of treatment. We have not pinpointed the optimal timeframe for any potential upregulation that may happen as compensation. Therefore, we have reworded our analysis in this section to not over-interpret the data in Figure 1 (lines 287-315) and to only focus on persistence.

Comment 10. Figure 4a. Is there any reason why the tumor volume decreases after day 16 in Vehicle control?

Response 10. Yes, the tumor volume decreases after day 16 in the Vehicle group because one mouse was euthanized due to the volume of its tumor exceeding the 2000 mm3  endpoint value as detailed in the methods section. We have added clarification in the results section (Lines 396-397) and the figure legend for Figure 4A (line 418).

Comment 11. Figure 5: Why you didn’t use SHP099 and Omni+shp099 for the in vivo study. Efficacy may vary during in vivo experiments. If you have data, put it in the supplementary section.

Response 11. We agree that in hindsight this would have been good to do. We did not use the OmiTram in the PKT mouse model shown in Figure 5 because the results from the subcutaneous implantation study Figure 4 suggested it would be less effective.

Comment 12. Figure 5. It would be good to show changes in the tumor microenvironment after treatment. At the very least, alterations in CD8 ( you can also show CD4, macrophages, NK cells, MDSCs etc.)

Response 12. There was little to no CD8 detected, as commonly observed for PDAC tumors. We have reported relative intensity for IBA1 (a myeloid marker) in Figure S6D and in lines 432-433. In short, we did observe a slight difference in IBA1 expression between the Omi and OmiTram groups, but no significant differences among the others.

Comment 13. Can you show the exact tumor volume or weight graph in Figure 5B and the animal weight graph in the supplementary?

Response 13. Yes, we have added this data. Please see Figure S6B, C.

Comment 14. Figure 5D shows no alterations in the stroma, which may be attributable to the PKT GEM model. Can you provide a similar staining (trichrome or Sirius red stain) for the in vivo study using the KPC cell line? Because KPC tumors have more stroma which resemble human PDAC?

Response 14. At the reviewer’s request, we have stained sections and added additional data to address this concern. Trichrome staining of the subcutaneous tumors shown in Figure 4 showed no significant differences among the groups. These data have been added to Figure S6A and lines 451-452.

Comment 15. Did you check the pERK and p-MEK levels in the in vivo tumor after therapy to ensure that the pathways were inhibited?

Response 15. We checked pERK and pAKT in endpoint tumors where possible to assess pathway activity after treatment (Figure S6E). At endpoint, pERK and pAKT levels were not uniform across the mice, which may be indicative of the development of resistance in some animals or just due to endpoint complications. Ideally, we should have taken a cohort of mice after 3 days of treatment to demonstrate in vivo efficacy of the inhibitors, but we do not have those samples.

Round 2

Reviewer 1 Report

Comments and Suggestions for Authors

Thanks to the authors for making clarifications and modifications to the manuscript.

Author Response

No additional suggestions were provided by the reviewer

Reviewer 2 Report

Comments and Suggestions for Authors

Dear Authors, 
I have carefully reviewed the corrected article and are pleased to note that my comments have been taken into account. The wording in the text has been adjusted in accordance with our comments, and the presented version looks clearer.

Author Response

(The authors gave the same response as above.)

Reviewer 3 Report

Comments and Suggestions for Authors

Dear Author,

 Thanks for the clarifications. I just have one more comment

1) Are you sure that IBA1 staining (Supplementary Figure 6D) can detect all myeloid populations in the PDAC tumor? Please add the reference in the manuscript if  IBA1 marker was used in any PDAC studies (IHC or IF) to illustrate the myeloid population. In most of the studies IBA1 marker was used as marker for microglia

Best Wishes

Author Response

Comment 1: Are you sure that IBA1 staining (Supplementary Figure 6D) can detect all myeloid populations in the PDAC tumor? Please add the reference in the manuscript if  IBA1 marker was used in any PDAC studies (IHC or IF) to illustrate the myeloid population. In most of the studies IBA1 marker was used as marker for microglia

Response 1: We appreciate this suggestion and have added a couple of select references to the manuscript text to support its selectivity in immune populations. To improve clarity, we have reworded the text to demonstrate IBA1 as a macrophage marker in PDAC [1-4] (lines 435-438). We also noted a small error in the data for Supplemental figure 6 for the sample number, which has been addressed in supplemental attachment as well as the associated manuscript text.

Although not included in the manuscript, we have provided the reviewer with further support for our use of IBA1 as a macrophage marker:

IBA1 (also called AIF1) has indeed been well-described as a marker of microglia [6, 9], but also has been broadly identified in other monocyte/macrophage subsets in tissues outside the nervous system via IF [5, 7, 10] and single-cell transcriptomic data [8, 11]. With respect to the pancreas specifically, IBA1 has been identified to be enriched in monocyte-derived cells (macrophages and potentially dendritic cells) from pancreatic cancer samples (via single-cell RNAseq, [11]) as well as in normal and diabetic pancreas samples via IHC and/or flow cytometry [10, 12]. IBA1 was additionally used by others in PDAC-associated single-cell RNAseq [1, 2] and histological [3, 4] studies to help identify macrophage populations.

Citations

  1. Peng, J., B. F. Sun, C. Y. Chen, J. Y. Zhou, Y. S. Chen, H. Chen, L. Liu, D. Huang, J. Jiang, G. S. Cui, Y. Yang, W. Wang, D. Guo, M. Dai, J. Guo, T. Zhang, Q. Liao, Y. Liu, Y. L. Zhao, D. L. Han, Y. Zhao, Y. G. Yang, and W. Wu. "Single-Cell Rna-Seq Highlights Intra-Tumoral Heterogeneity and Malignant Progression in Pancreatic Ductal Adenocarcinoma." Cell Res 29, no. 9 (2019): 725-38.
  2. Yang, K., T. Yang, J. Yu, F. Li, and X. Zhao. "Integrated Transcriptional Analysis Reveals Macrophage Heterogeneity and Macrophage-Tumor Cell Interactions in the Progression of Pancreatic Ductal Adenocarcinoma." BMC Cancer 23, no. 1 (2023): 199.
  3. Bu, L., A. Yonemura, N. Yasuda-Yoshihara, T. Uchihara, G. Ismagulov, S. Takasugi, T. Yasuda, Y. Okamoto, F. Kitamura, T. Akiyama, K. Arima, R. Itoyama, J. Zhang, L. Fu, X. Hu, F. Wei, Y. Arima, T. Moroishi, K. Nishiyama, G. Sheng, T. Mukunoki, J. Otani, H. Baba, and T. Ishimoto. "Tumor Microenvironmental 15-Pgdh Depletion Promotes Fibrotic Tumor Formation and Angiogenesis in Pancreatic Cancer." Cancer Sci 113, no. 10 (2022): 3579-92.
  4. Panebianco, C., A. Villani, F. Pisati, F. Orsenigo, M. Ulaszewska, T. P. Latiano, A. Potenza, A. Andolfo, F. Terracciano, C. Tripodo, F. Perri, and V. Pazienza. "Butyrate, a Postbiotic of Intestinal Bacteria, Affects Pancreatic Cancer and Gemcitabine Response in in Vitro and in Vivo Models." Biomed Pharmacother 151 (2022): 113163.
  5. Köhler, C. "Allograft Inflammatory Factor-1/Ionized Calcium-Binding Adapter Molecule 1 Is Specifically Expressed by Most Subpopulations of Macrophages and Spermatids in Testis." Cell Tissue Res 330, no. 2 (2007): 291-302.
  6. Sasaki, Y., K. Ohsawa, H. Kanazawa, S. Kohsaka, and Y. Imai. "Iba1 Is an Actin-Cross-Linking Protein in Macrophages/Microglia." Biochem Biophys Res Commun 286, no. 2 (2001): 292-7.
  7. Donovan, K. M., M. R. Leidinger, L. P. McQuillen, J. A. Goeken, C. M. Hogan, S. C. Harwani, H. A. Flaherty, and D. K. Meyerholz. "Allograft Inflammatory Factor 1 as an Immunohistochemical Marker for Macrophages in Multiple Tissues and Laboratory Animal Species." Comp Med 68, no. 5 (2018): 341-48.
  8. Karlsson, M., C. Zhang, L. Méar, W. Zhong, A. Digre, B. Katona, E. Sjöstedt, L. Butler, J. Odeberg, P. Dusart, F. Edfors, P. Oksvold, K. von Feilitzen, M. Zwahlen, M. Arif, O. Altay, X. Li, M. Ozcan, A. Mardinoglu, L. Fagerberg, J. Mulder, Y. Luo, F. Ponten, M. Uhlén, and C. Lindskog. "A Single-Cell Type Transcriptomics Map of Human Tissues." Science Advances 7, no. 31 (2021).
  9. Pfrieger, F. W., and M. Slezak. "Genetic Approaches to Study Glial Cells in the Rodent Brain." Glia 60, no. 5 (2012): 681-701.
  10. Chen, Z. W., B. Ahren, C. G. Ostenson, A. Cintra, T. Bergman, C. Möller, K. Fuxe, V. Mutt, H. Jörnvall, and S. Efendic. "Identification, Isolation, and Characterization of Daintain (Allograft Inflammatory Factor 1), a Macrophage Polypeptide with Effects on Insulin Secretion and Abundantly Present in the Pancreas of Prediabetic Bb Rats." Proc Natl Acad Sci U S A 94, no. 25 (1997): 13879-84.
  11. Liu, X., D. Zhang, J. Hu, S. Xu, C. Xu, and Y. Shen. "Allograft Inflammatory Factor 1 Is a Potential Diagnostic, Immunological, and Prognostic Biomarker in Pan-Cancer." Aging (Albany NY) 15, no. 7 (2023): 2582-609.
  12. Elizondo, D. M., N. Z. Brandy, R. L. da Silva, T. R. de Moura, and M. W. Lipscomb. "Allograft Inflammatory Factor-1 in Myeloid Cells Drives Autoimmunity in Type 1 Diabetes." Jci Insight 5, no. 10 (2020).